# Regulation on Aggregation Behavior and In Vitro Digestibility of Phytic Acid–Whey Protein Isolate Complexes: Effects of Heating, pH and Phytic Acid Levels

**DOI:** 10.3390/foods13213491

**Published:** 2024-10-31

**Authors:** Yaqiong Pei, Ziyu Deng, Bin Li

**Affiliations:** 1College of Food Science and Technology, Wuhan Business University, Wuhan 430056, China; 2College of Food Science and Technology, Huazhong Agricultural University, Wuhan 430070, China

**Keywords:** phytic acid, interaction, aggregation behavior, hydrolysis degree

## Abstract

The impact of heat treatment, pH and phytic acid (PA) concentration on the aggregation behavior and digestibility of whey protein isolate (WPI) was investigated. The experimental results indicated that below the isoelectric point of WPI, heat treatment and elevated PA levels significantly increased turbidity and particle size, leading to the aggregation of WPI molecules. No new chemical bonds were formed and the thermodynamic parameters ΔH < 0, ΔS > 0 and ΔG < 0 suggested that the interaction between PA and WPI was primarily a spontaneous electrostatic interaction driven by enthalpy. After the small intestine stage, increasing phytic acid levels resulted in a significant decrease in hydrolysis degree from 16.2 ± 1.5% (PA0) to 10.9 ± 1.4% (0.5% PA). Conversely, above isoelectric point of WPI, there was no significant correlation between the presence of PA and the aggregation behavior or digestion characteristics of WPI. These results were attributed to steric hindrance caused by PA-WPI condensates, which prevented protease binding to hydrolysis sites on WPI. In summary, the effect of PA on protein aggregation behavior and digestive characteristics was not simply dependent on its presence but largely on the aggregation degree of PA-WPI induced by heat treatment, pH and PA concentration. The findings obtained here suggested that phytic acid may be utilized as an agent to modulate the digestion characteristics of proteins according to production requirements. Additionally, the agglomerates formed by heating phytic acid and protein below the isoelectric point could also be utilized for nutrient delivery.

## 1. Introduction

The interaction between proteins and plant secondary metabolites significantly influences the structure, functional properties and bioavailability of proteins [1]. Given that protein is one of the most essential nutrients for the human body, various substances, including tea polyphenols [2], tannic acid [3], phytate and other phenolic acid substances [4], have long been identified as antinutrients.

Phytic acid, in particular, is naturally present in relatively high concentrations (0.5–5% *w*/*w*) in certain cereals, legumes, and oil seeds [5]. Due to its unique chemical structure, which comprises six phosphate groups, phytic acid carries a negative charge across the pH range typically encountered in foods [6]. This characteristic enables it to electrostatically interact with positively charged amino acid residues (such as arginine, lysine, and histidine) in proteins, leading to the formation of complexes [7]. To mitigate the adverse effects of phytic acid in food, numerous strategies have been employed, ranging from crop breeding to food processing techniques, including soaking, germination, fermentation, extrusion, and the addition of exogenous phytase [8,9,10]. However, it is important to recognize that, beyond its negative implications, dietary phytic acid also offers beneficial health effects, such as protection against various cancers, heart-related diseases, diabetes mellitus, and renal stones [11,12,13]. Therefore, it may be inappropriate to advocate for a unilateral reduction in phytic acid in food.

Phytic acid reduces the rate of protein digestion and absorption primarily due to its strong negative charge, which facilitates the formation of binary or ternary complexes with proteins and mineral ions [5,14]. These complexes are resistant to hydrolysis by proteases, leading to a diminished degree of hydrolysis during protein digestion [15,16]. However, high ionic strength enhances the digestibility of protein by diminishing the interactions between proteins and phytic acid, thereby inhibiting the formation of phytic acid–protein complexes. Additionally, heating treatment improves protein digestibility by exposing various sites on the protein molecules, rendering them more accessible to enzymes [4]. Furthermore, while phytic acid inhibits protein digestion mainly through a complex formation, it does not alter the structure or activity of proteases [17,18]. In summary, the impact of phytic acid on the digestive properties of proteins can be regulated by controlling the interaction behaviors between phytic acid and proteins, which are significantly influenced by solution parameters, such as component composition, concentration ratio, pH, and ionic strength [19,20].

Furthermore, due to the complexity of the human digestive tract environment (gastric environment, 1.0 < pH < 3.0; intestinal environment, 6 < pH < 7.5) [21], the size and morphology of PA–protein complexes may change during gastrointestinal digestion, which could ultimately affect the hydrolytic degree of proteins. Therefore, it is particularly important to explore the relationship between the interaction behavior and micromorphology of phytic acid–protein complexes, the changes in these complexes during gastrointestinal digestion, the digestibility of proteins and the related underlying mechanisms.

In the current study, we investigated the interaction behavior of phytic acid (PA) with whey protein isolate (WPI) using the kinetic turbidimetric method and isothermal titration calorimetry (ITC) under varying pH and temperature conditions. The influences of heating treatment, pH values and PA levels on the microstructure, along with their relationship to the in vitro digestibility of the PA-WPI complexes, were also evaluated. The findings may guide the rational application of phytic acid as an excipient ingredient in functional foods and provide reference information for the processing and production of natural phytic acid-containing food raw materials.

## 2. Materials and Methods

### 2.1. Materials

Powdered whey protein isolate (WPI, dry basis content 97.6%; BiPro JE-099-2-420) was obtained from Davisco Foods International Inc. (Le Sueur, MN, USA) Phytic acid (PA, 98%; Lot #BCBR3133V) derived from rice, along with pepsin from porcine gastric mucosa (250 units/mg) and trypsin (100−400 units/mg) were sourced from Sigma-Aldrich (St. Louis, MO, USA). All other reagents used were of analytical grade. All samples were prepared using double-distilled water produced via a laboratory-grade water purification system.

### 2.2. PA-WPI Mixed Solution Phase Behavior Using Turbidimetric Titration

Initially, a WPI stock solution was prepared by dispersing 5.0% *w*/*w* WPI powder in water with constant stirring (600 rpm) at 25 °C for 2 h. This solution was subsequently stored at 4 °C for 12 h to ensure the complete dissolution of the WPI molecules.

The PA-WPI mixed solution was prepared by diluting the WPI stock solution (final concentration fixed at 0.1% *w*/*w*) and incorporating varying levels of PA (0, 0.001, 0.0025, 0.005, 0.0085 and 0.01% *w*/*w*). A pH–turbidity curve was obtained by titrating HCl (0.5, 1 M) or NaOH (0.1, 0.5 M) to adjust the pH of the PA-WPI mixed solution (from pH 7.0 to pH 2.0) while monitoring its transmittance (T%) using a UV spectrophotometer (Mepuda Instrument, Shanghai, China) at a wavelength of 600 nm. The turbidity of the solution was expressed as 100-T%.

The Zetasizer Nano ZS (Malvern Instruments, Malvern, UK) was utilized to determine the zeta potential of the PA-WPI mixed solution, following a previously determined method [22]. Samples were loaded into an appropriate cuvette (folded capillary zeta cell DTS1070) (Malvern Instruments, Malvern, UK), and the zeta potential was tested by measuring the direction and velocity of the complexes in the applied electric field. The Smoluchowski Equation (F(ka) 1.5) was employed to calculate the zeta potential. Measurements were conducted on three freshly prepared samples, with three readings taken per sample.

### 2.3. Isothermal Titration Calorimetry (ITC)

Energetic flow and binding parameters during the interaction between phytic acid and β-Lg (β-lactoglobulin) at varying pH values and ionic temperatures were investigated using isothermal titration calorimetry (ITC; ITC200, Malvern Instruments, UK). Initially, β-Lg solution (0.2 mM) was dialyzed in a sodium acetate–acetic acid buffer solution (20 mM) at pH 3.5, pH 4.5 and pH 5.6 for 24 h. The same buffer system at different pH values was employed to minimize the influence of the solution background. A 4 mM phytic acid (PA) solution was prepared using the sodium acetate–acetic acid buffer solution following the dialysis of proteins (at pH 3.5, pH 4.5 and pH 5.6). The sample cell and reference cell were each loaded with 200 µL of β-Lg solution and sodium acetate–acetic acid buffer solution following the dialysis of β-Lg. Additionally, 40 µL of PA solution was loaded into an injection syringe. Titration was conducted by performing an initial injection of 0.5 µL, followed by 19 successive injections of 2 µL each, with 90 s intervals between injections. The system was stirred at 500 rpm. The thermodynamic parameters, including the stoichiometric ratio (N), binding constant (K), entropy (ΔS), enthalpy (ΔH), and Gibbs free energy (ΔG = ΔH − TΔS), were derived using the ITC origin analysis software 1.0 with the ‘single point combined with (one set of sites)’ model to fit the titration results, discarding the initial 0.5 µL injection.

### 2.4. Fourier Transform Infrared (FTIR) Analysis

FT-IR spectra of the sample were obtained by scanning from 4000 cm^−1^ to 400 cm^−1^ at 25 °C (4100 series, Jasco Inc., Easton, PA, USA). Prior to scanning, the freeze-dried samples were mixed with KBr at a ratio of 1:50 and compressed into thin round sheets. A blank KBr sheet was scanned as a background before each sample scan.

### 2.5. PA-WPI Mixed Samples Preparation and Characterization

Samples preparation: PA-WPI mixed samples were prepared by diluting the WPI stock solution (with a final concentration fixed at 1.0% *w*/*w*) and adding varying levels of PA (0, 0.05, 0.1, 0.2 and 0.5% *w*/*w*). The pH of the PA-WPI mixed samples was adjusted to 7.0 and 3.5 using 0.5 M HCl or 0.5 M NaOH. The resulting solution was then heated to 90 °C in a water bath for 30 min to induce the denaturation of WPI, followed by cooling in tap water.

Particle size measurements: According to a previously described method, a Mastersizer was employed to measure the particle size of samples ranging from 0.1 to 1000 μm, while dynamic light scattering (DLS) was utilized for samples with particle sizes between 0.3 and 6000 nm [23]. Initially, the Malvern Mastersizer was chosen for measuring the particle size of the PA-WPI complex due to its suitability for relatively larger particles. However, potential dilution and agitation during testing may have caused a partial disruption of the flocculated PA-WPI particles, resulting in undetectable results. Consequently, the dynamic light scattering (DLS) method was employed to measure the particle size of the PA-WPI complexes. The mean particle size diameter was determined using a Zetasizer Nano ZS (Malvern Instruments, UK) at 25 °C, calculated from the translational diffusion coefficient using the Stokes–Einstein equation.

Microstructure analysis: The microstructures of the samples were measured using laser scanning confocal microscopy (LSCM) equipped with a 40× objective and a 10× eyepiece lens (Nikon D-Eclipse C1 80i, Nikon, Tokyo, Japan). Prior to observation, 1 mL of the samples was combined with 20 µL of FITC solution (1 mg/mL in ethanol, a protein stain). A drop of sample was placed onto a glass slide and subsequently covered with a coverslip. Microstructure images were obtained and processed using image analysis software (NIS-Elements, Nikon).

### 2.6. In Vitro Digestion of PA-WPI Mixtures

The in vitro digestibility of whey protein isolate (WPI) within PA-WPI mixtures was evaluated using a method described previously [24,25], with slight modifications. A 10 g sample of the PA-WPI mixed solution was added to 50 g of simulated gastric fluid (SGF: 2 g/L NaCl, 7 mL/L of 12 M HCl, and 3.2 g/L of pepsin) and the pH was adjusted to 2.5 using 1.0 M HCl or NaOH. The mixture was then incubated in an incubator shaker (100 rpm, 37 ℃) for 2 h. Subsequently, the sample resulting from the gastric stage was adjusted to pH 7.0 using 1.0 M NaOH. Following this, 0.09 g of trypsin (dissolved in 2.5 mL of 10 mM phosphate buffer, pH 7.0) was added, and the sample was subjected to the small intestinal stage (2 h, 100 rpm, 37 °C). During the digestion process, a 5 g sample was periodically collected and, heated in a boiling water bath for 5 min to inactivate the enzyme, and then stored at 4 °C or freeze-dried for a further determination of the degree of hydrolysis and SDS-PAGE analysis of the PA-WPI mixtures.

### 2.7. The Degree of Hydrolysis Determination

The degree of hydrolysis (DH) of whey protein isolate (WPI) within PA-WPI mixtures was measured using the π-phthaldialdehyde (OPA) method [24,26]. Samples at different digestion times were adjusted to pH 7.0 and centrifuged at 10,000 rpm for 30 min using a 10K ultrafiltration centrifuge tube (Merk Millipore, Bedford, MA, USA) to minimize interference from protein molecules. A volume of 0.2 mL of the centrifuge sample was mixed with 1.5 mL OPA solution (OPA solution was prepared by completely dissolving 3.18 g of sodium tetraborate and 0.1 g of SDS in 80 mL of double-distilled water. Subsequently, 2 mL of anhydrous ethanol containing 0.08 g of phthalaldehyde and 0.088 g of dithiothreitol was added, and the mixture was adjusted to a final volume of 100 mL. The solution was then filtered through a 0.45 µm membrane) and allowed to stand for exactly 3 min before measurement at 340 nm using a UV spectrophotometer. The concentration of serine was calculated according to a standard curve generated from serine concentrations of 0, 25, 50, 150, 200 and 300 mg/L. Consequently, the DH of WPI within PA-WPI mixtures was calculated using the following formula:DH%=mS-NH2×0.9516100/P−βαHtotal×100%

*m_S-NH2_* represents the concentration of serine as determined by the standard curve (mg/L). The value 0.9516 denotes the molar equivalent of serine (meqv/L); *P* indicates the protein concentration in the sample (g/L); and *α* (1.0), *β* (0.4) and *H_total_* (8.8) are constant values for whey protein as reported in the literature [26].

### 2.8. SDS-Page Analysis of Peptic Hydrolysis

The hydrolysis of protein and the production of polypeptides were analyzed using SDS-PAGE, as described by Ye et al. [27], with appropriate modifications. The electrode buffer composition consisted of 0.05 M Tris, 0.384 M glycine and 0.1% *w*/*v* SDS at pH 8.39. The sample buffer composition included 0.2 M Tris-HCl, 2.0% (*v*/*v*) mercaptoethanol, 5.0% (*v*/*v*) glycerol, 1.0 % (*w*/*v*) SDS, and 0.025% (*w*/*v*) bromophenol blue. The concentrations of acrylamide in the stacking gel and resolving gel were 5% and 15%, respectively. The freeze-dried sample, obtained after the small intestine stage, was dissolved in sample buffer (0.3 mg/mL), thoroughly mixed using a vortex, boiled in a water bath for 5 min, cooled to room temperature, and centrifuged for 3 min at 10,000 rpm. Subsequently, 20 µL of the supernatant was loaded into stacking gel and electrophoresis was conducted at 80 mA. Following this, the gel was stained for 3 h in a dye solution (Blue R-250) and then decolorized using a methanol–acetic acid solution three time. Electrophoretic profiles were employed to assess the changes in whey protein isolate after digestion.

### 2.9. Statistical Analysis

All experiments were conducted in triplicate, and the results were expressed as means ± standard deviations (SDs). The data and figures were processed using Excel 2021 and origin 8.5. Differences between samples were calculated using the least significant difference (LSD) method (SPSS 22.0, *p* < 0.05).

## 3. Results and Discussion

### 3.1. Characterization of the Interaction Between WPI and PA

#### 3.1.1. Turbidity and Zeta Potential of PA-WPI Solutions

Initially, the effects of pH (ranging from 2.0 to 7.0) and PA concentration (from 0 to 0.01% *w*/*w*) on the turbidity and zeta potential of PA-WPI solutions were investigated at a WPI concentration of 0.1% *w*/*w* (Figure 1). In general, the turbidity of PA-WPI solutions increased initially before decreasing with a reduction in pH value (Figure 1A). The turbidity of PA-WPI mixtures with varying PA levels remained relatively constant (close to zero) at pH values above 5.3. This phenomenon occurred because both PA and WPI were negatively charged at pH > 5.3, resulting in electrostatic repulsion and thus a turbidity close to zero. The turbidity curves exhibited a significant increase when the pH value dropped below 5.3, peaking at 95.9% turbidity at pH 5.0 (Figure 1A), which closely approached the isoelectric point of the whey proteins. According to Li et al. [28], a strong aggregation of the complexes occurred around the isoelectric points of proteins. The occurrence of these phenomena was attributed to the electrostatic interaction between the local positive charge of the WPI molecule and the negative charge of the PA molecule at the pH value corresponding to the peak turbidity. Subsequently, turbidity declined at lower pH values due to the re-dissolution of the precipitate (Figure 1B). The observed outcomes were attributed to the protonation and deprotonation of PA molecules at varying pH values [29,30]. Our previous work demonstrated that the charge of PA exhibited pH-dependent behavior, displaying a highly negative charge at elevated pH levels and an almost neutral charge at very low pH levels [19]. Consequently, at a low pH value, there was a reduction in charge magnitude, which weakened the interaction between PA and WPI [7,31].

A significant increase in maximum turbidity at pH 5.0 was observed as the concentration of PA increased from 0 (turbidity 13.4%) to 0.01% (turbidity 95.9%). Furthermore, the pH range of the WPI-PA interaction was expanded (Figure 1A). The presence of higher concentrations of PA in the solution resulted in an enhanced release of counterions, thereby increasing the electrostatic neutralization effect and promoting coacervation, which ultimately led to an elevation in turbidity. This trend was evident in the decrease in the magnitude of the zeta potential below the isoelectric point of WPI (Figure 1B) [31].

#### 3.1.2. FTIR Analysis

The interaction between PA and WPI was characterized by Fourier transform infrared (FT-IR) spectroscopy (Figure 2). The spectra of PA exhibited four characteristic bands: the HPO_4_^2-^ stretching vibration peak at 1648 cm^−1^, the P=O stretching vibration peak at 1170 cm^−1^, the C-O-P stretching vibration peak at 1124 cm^−1^, and the PO_4_^3−^ stretching vibration at 970 cm^−1^ [32,33]. The WPI spectra displayed two prominent characteristic peaks at 1648 cm^−1^ (representing amide I, indicative of C=O stretching) and 1533cm^−1^ (corresponding to amide II, arising from N-H bending and C-N stretching). The absorption peaks demonstrated a high degree of similarity among WPI (pH 7.0), WPI (pH 3.5), and PA-WPI (pH 7.0), indicating that the treatments had no discernible impact on the structural integrity of WPI.

It was observed that the absorption peaks were analogous among WPI (pH 7.0), WPI (pH 3.5), and PA-WPI (pH 7.0), suggesting that the presence of the PA molecule at pH 7.0 did not affect the structure of WPI. In contrast to the spectra of WPI, the spectra of PA-WPI (pH 3.5) exhibited characteristic peaks corresponding to the PA phosphate group, indicating an electrostatic interaction between PA and WPI [34]. Furthermore, no additional peaks were observed in the spectra of PA, WPI, and PA-WPI, suggesting that no new chemical bonds were formed. Consequently, it can be inferred that the chemical structure of WPI remained unaffected by the presence of PA.

#### 3.1.3. Isothermal Titration Calorimetry (ITC) Analysis

To further investigate the interaction between PA and WPI, ITC was utilized to characterize the thermodynamics of complex formation.

As depicted in Figure 3, at pH 5.6, there was a negligible heat flow change except for the minimal heat flow attributed to the diffusion of small molecules during the titration. This observation suggested that no binding occurred between PA and the protein due to their negative charges at pH levels above the isoelectric point of WPI. However, the interaction of PA and the protein at pH 3.5 and 4.0 exhibited exothermic combinations, indicating spontaneous reactions. The binding curves displayed a characteristic S-shape, consistent with typical protein–small molecule interactions [4,35]. The combined heat flow of PA and β-Lg consistently decreased as the number of titrations increased, eventually reaching a steady state, suggesting that the combination of PA and β-Lg became saturated [16]. Additionally, saturation was achieved at the 11th drop at pH 3.5, while it occurred at the 10th drop at pH 4.0. Notably, a higher heat flow was observed at pH 3.5 compared to pH 4.0, indicating a stronger interaction force between PA and β-Lg and consequently a higher degree of binding at pH 3.5. This could be attributed to the elevated charge density of β-Lg under acidic conditions (pH 3.5).

The thermodynamic parameters of PA and β-Lg upon their combination at pH 3.5 and at different temperatures are presented in Figure 4. These parameters included the stoichiometric binding ratio (N), binding constant (K), enthalpy change (ΔH), entropy change (ΔS), and Gibbs free energy change (ΔG). The stoichiometric binding ratio (N) remained consistent across all samples at the same pH as the temperature increased from 15 °C to 45 °C, indicating that the N value between PA and β-Lg was unaffected by temperature [4]. The value of K generally increased with rising temperature, suggesting an enhancement in the hydrophobic interaction force among protein molecules at elevated temperatures. The negative change in enthalpy (ΔH < 0), positive change in entropy (ΔS > 0), and negative change in free energy (ΔG < 0) collectively indicated that the binding process between PA and WPI was a spontaneous electrostatic interaction, primarily governed by enthalpy [16].

### 3.2. Aggregation Behavior PA-WPI Mixture

The microstructure and particle size changes in the PA-WPI mixture before and after heat treatment are illustrated in Figure 5. The microstructure of PA-WPI mixtures remained uniformly distributed in both the heated and non-heated groups at pH 7.0, with a particle size of 90 nm, and was unaffected by PA concentration. As discussed in Section 3.1.1, both WPI and PA exhibited negative charges under this specific pH condition, resulting in electrostatic repulsion that prevented aggregation.

In contrast, the unheated samples exhibited varying degrees of aggregation at pH 3.5. The degree of aggregation and particle size increased proportionally with the concentration of PA due to the enhanced electrostatic interaction between WPI and PA. Additionally, the heated samples demonstrated a higher level of aggregation and larger particle size compared to the non-heated samples. This phenomenon could be attributed to the alteration of protein conformation induced by heat treatment, which resulted in decreased protein homogeneity and solubility, ultimately leading to protein aggregation [36,37]. However, the size observed in the microstructure images appeared larger than the particle size depicted in Figure 5B, likely due to the PA-WPI complex forming an amorphous flocculate rather than a rigid solid particle. Additionally, the particle size measured using laser scanning confocal microscopy (LSCM) was reflective of the local field of view, whereas the results from dynamic light scattering (DLS) provided an average of the overall particle size. Although discrepancies in particle size were noted between the two testing methods, both indicated the aggregation of PA and WPI below the isoelectric point, with heat treatment further intensifying this aggregation.

The effects of heat treatment, pH and PA concentration on the microstructure and aggregation behavior of proteins during processing will undoubtedly influence the digestion behavior and bioavailability of proteins.

### 3.3. In Vitro Digestion Behavior

#### 3.3.1. Effect of Heat Treatment and pH

The impact of heat treatment and pH on the degree of protein hydrolysis is illustrated in Figure 6.

Heat treatment: Following a simulated gastrointestinal condition (SGC) for 4 h, the heated samples exhibited a significantly higher degree of protein hydrolysis (11.5 ± 0.5% to 16.7 ± 0.4%) compared to the non-heated samples (<2.5%). This result indicated that the WPI in the non-heated group were not easily digested and hydrolyzed. The SDS-PAGE image also revealed that the majority of protein molecules in the heated group had undergone hydrolysis, resulting in the formation of peptides and hydrolysis products, with only a limited presence of intact protein molecules. In contrast, the non-heated samples exhibited distinct bands corresponding to intact WIP molecules (main components: β-lactoglobulin, β-Lg; α-lactalbumin, α-La), demonstrating reduced susceptibility to enzymatic digestion. The structural stability of β-Lg under low pH conditions has been previously reported, wherein the majority of its hydrophobic amino acid groups were tightly enclosed within its hydrophobic core region, exhibiting limited accessibility [27,38]. However, these amino acids serve as susceptible sites for protease cleavage; thus, non-heated WPI demonstrates resistance to enzymatic digestion. Conversely, heat treatment could induce conformational changes in proteins, thereby exposing enzyme binding sites and increasing the susceptibility of WPI to protease hydrolysis [39,40]. Consequently, the degree of the hydrolysis of WPI in heated samples was significantly higher compared to that in non-heated samples.

pH: The degree of hydrolysis in the heated samples of WPI at pH 7.0(16.7 ± 0.4%), WPI at pH 3.5 (15.9 ± 0.8%) and PA-WPI at pH 7.0 (14.9 ± 0.8%) exhibited no significant difference. As mentioned in Section 3.2, the heat treatment of the PA-WPI mixture at pH 7.0 did not affect the thermal aggregation behavior of WPI, resulting in no significant impact on its hydrolysis degree. Consequently, the slightly lower final hydrolysis degree observed might be attributed to the aggregation between PA and WPI in the acidic gastric fluid environment [21]. A notable reduction in the hydrolysis degree of PA-WPI at pH 3.5 (10.5 ± 0.6%) was observed, attributed to the formation of PA-WPI complex condensates at this pH level. Furthermore, heat treatment enhanced the degree of aggregation and impeded pepsin binding to the proteolytic site, thereby significantly diminishing the extent of hydrolysis. This result revealed that PA had no significant effect on pepsin activity [17].

#### 3.3.2. Effect of PA Concentration

According to the findings in Section 3.2, the effects of PA concentration on protein digestion behavior discussed below were all samples following heat treatment. Overall, the hydrolysis degree of WPI in all PA-WPI mixtures consistently increased with a prolonged digestion time (Figure 7).

The hydrolysis degree of PA-WPI heat at pH 7.0 showed no significant difference during the initial 30 min of the gastric digestion stage. However, starting from 1 h, a slight decrease in the hydrolysis degree was noted with increasing PA concentration at the same digestion time point. At the conclusion of the gastric digestion phase, PA-WPI with a high PA concentration (0.5%) exhibited a slightly lower hydrolysis degree (11.3 ± 0.4%) compared to the PA0 sample (12.7 ± 0.3%). As previously mentioned, this phenomenon might be attributed to the interaction between PA and WPI particles in the acidic environment of gastric fluid, resulting in a certain degree of flocculation and a reduction in the hydrolysis degree. Upon entering the small intestine digestion stage, the degree of hydrolysis for all samples continued to increase (Figure 7A). The results indicated that the hydrolysis degree of PA-WPI at a high PA concentration (0.5%) was only slightly lower (15.6 ± 0.6%) than that of the PA0 sample (16.8 ± 0.3%), with no significant difference observed (Figure 7B). Therefore, in the non-aggregative state of the PA-WPI mixture, the presence of PA did not significantly impact the digestibility of WPI. The validity of this assertion was further substantiated by subsequent studies, which demonstrated that the digestion of the PA–protein mixture could be enhanced by inhibiting PA–protein binding through the addition of salt ions [4,41].

For samples of PA-WPI heated at pH 3.5 (Figure 7B), the hydrolysis degree of WPI with varying PA concentrations exhibited notable variations. At the end of the gastric digestion phase, the degree of hydrolysis decreased from 12.1 ± 0.3% (PA0) to 6.9 ± 0.4% (PA 0.5%). The rate of protein hydrolysis was contingent upon the accessibility of the enzyme to the cleavage sites within the protein molecule. As mentioned in Section 3.2, the flocculation degree of PA-WPI heated at pH 3.5 demonstrated a positive correlation with the concentration of PA, which was accompanied by an increase in particle size. Most enzyme cleavage sites within the WPI molecules were sequestered within the flocs, rendering them inaccessible to proteases and resulting in reduced protein hydrolysis. Consequently, following digestion in the small intestine, an increase in PA concentration led to a significant reduction in the rate of hydrolysis: 16.2 ± 0.8% (PA0) > 16.0 ± 0.6%(PA0.05%) > 13.1 ± 0.6% (PA0.1%) > 10.5 ± 0.6% (PA 0.2%) ≈ 10.9±0.9% (PA 0 5%) (Figure 7C).

The SDS-PAGE results (Figure 8) indicated that the presence of PA did not significantly inhibit the digestion and hydrolysis characteristics of WPI at pH 7.0, as only a minimal amount of intact β-Lg molecules were observed in samples with varying PA concentrations. This suggested that the majority of proteins were effectively hydrolyzed [40]. Notably, in the PA-WPI at pH 3.5, the protein bands of the PA0 sample were significantly weakened, indicating that most protein molecules in the PA0 sample were digested and hydrolyzed. In contrast, the protein band signal of samples with varying PA levels was significantly higher than that of the PA0 sample, suggesting that the hydrolysis of PA-WPI was inhibited to some extent. These results were consistent with the observed degree of hydrolysis.

In summary, phytic acid (PA) could reduce the digestibility of whey protein isolate (WPI) through two primary mechanisms: (1) During the digestion process, PA induced protein aggregation in the gastric environment, and the steric hindrance effect limited the interaction between pepsin and WPI, thereby reducing the extent of digestion and hydrolysis. (2) During processing, PA interacted with WPI below the isoelectric point of proteins to form a complex condensate, altering the polar microenvironment of specific amino acids in WPI and effectively hiding its enzymatic hydrolysis site within the aggregate (or hydrophobic core region). Following heat treatment, the degree of aggregation of the PA-WPI complex increased, making dissociation in the gastric digestive environment more challenging. The steric hindrance further complicated the binding of protease to the enzymatic hydrolysis sites of WPI, resulting in a decreased degree of WPI digestion and hydrolysis. The first mechanism resulted in a slight reduction in the degree of protein hydrolysis without a significant difference, while the second mechanism led to a significant reduction, representing the primary reason for PA’s inhibition of protein digestibility.

## 4. Conclusions

In conclusion, this study demonstrated that key factors such as heat treatment, pH and phytic acid (PA) concentration significantly influence the aggregation behavior and in vitro digestibility of whey protein isolate (WPI). Isothermal titration calorimetry (ITC) and Fourier transform infrared (FTIR) analyses indicated that the interaction mechanism between PA and WPI was characterized by enthalpy-driven physical electrostatic interactions, without alterations in chemical groups or the formation of new chemical bonds. Above the isoelectric point of WPI, both WPI and PA existed in a co-soluble state, where PA did not affect the thermal aggregation behavior of WPI or the exposure of enzymatic hydrolysis sites. There was no significant difference in the degree of digestion of PA-WPI mixtures at varying PA concentrations. Conversely, below the isoelectric point, the turbidity of the PA-WPI mixed system increased, as PA and WPI formed an insoluble complex. Additionally, the degree of aggregation of this complex increased with higher PA concentrations, and heat treatment further enhanced this aggregation, resulting in a larger particle size. Consequently, the polar microenvironment of the protein hydrolysis site changes, effectively wrapping or hiding it within the hydrophobic region. This steric hindrance inhibited protease binding to the enzymatic hydrolysis sites of WPI, leading to a decreased degree of WPI digestion.

Overall, our study suggests that the digestive properties of protein products can be regulated by controlling the aggregation behavior of phytate–protein mixed systems: protein digestibility is not affected when PA-WPI is in a co-soluble state; however, when PA-WPI is aggregated, it can be utilized in the development of nutrient encapsulation and delivery systems. 

## Figures and Tables

**Figure 1 foods-13-03491-f001:**
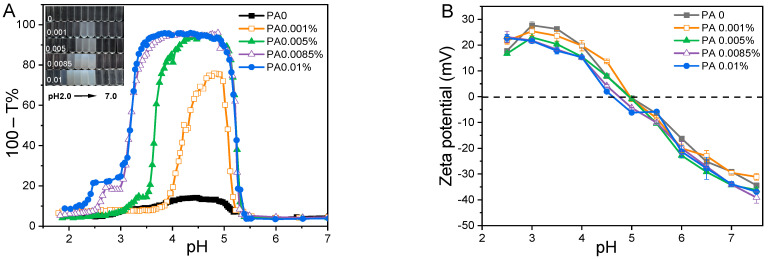
Turbidity curves (**A**) and zeta potential (**B**) of PA-WPI mixture solution as a function of pH at various PA concentrations (from 0 to 0.01% *w*/*w*).

**Figure 2 foods-13-03491-f002:**
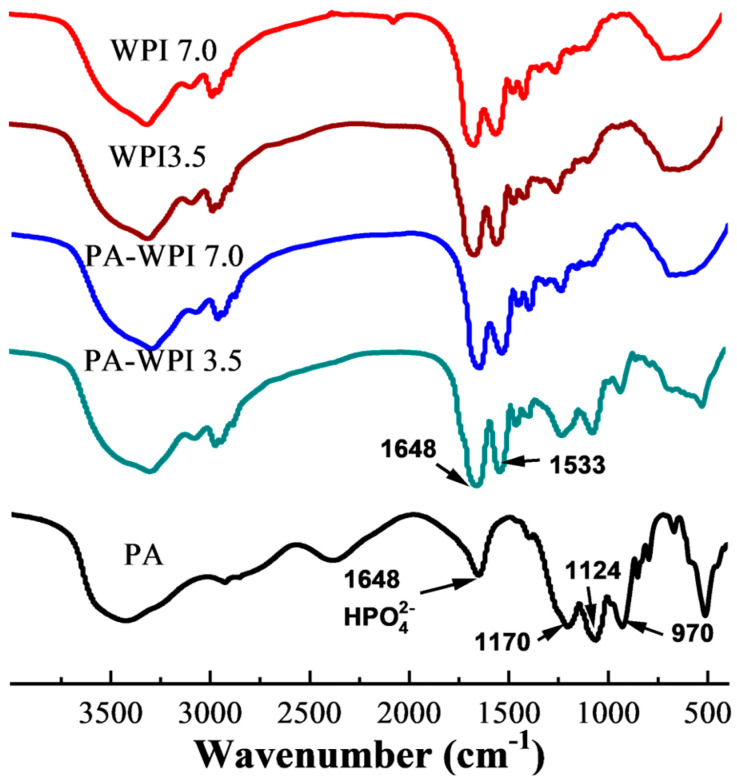
FTIR spectra of WPI and PA-WPI mixtures containing 0.2% *w*/*w* PA and 1.0% *w*/*w* WPI at pH 3.5 and pH 7.0.

**Figure 3 foods-13-03491-f003:**
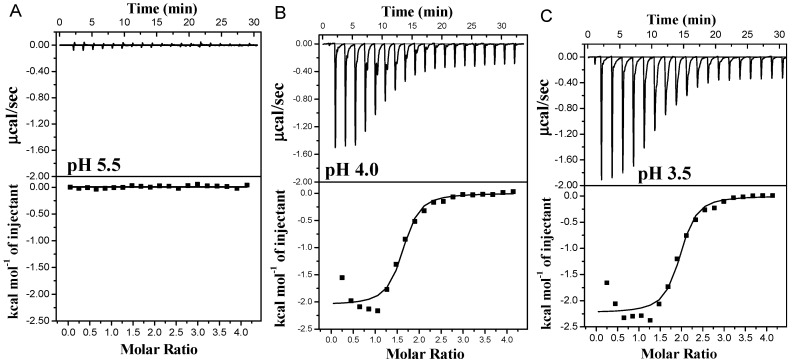
Thermograms and binding isotherms corresponding to the titration of PA with β-Lg at varying pH values: pH 5.5 (**A**), pH 4.0 (**B**) and pH 3.5 (**C**).

**Figure 4 foods-13-03491-f004:**
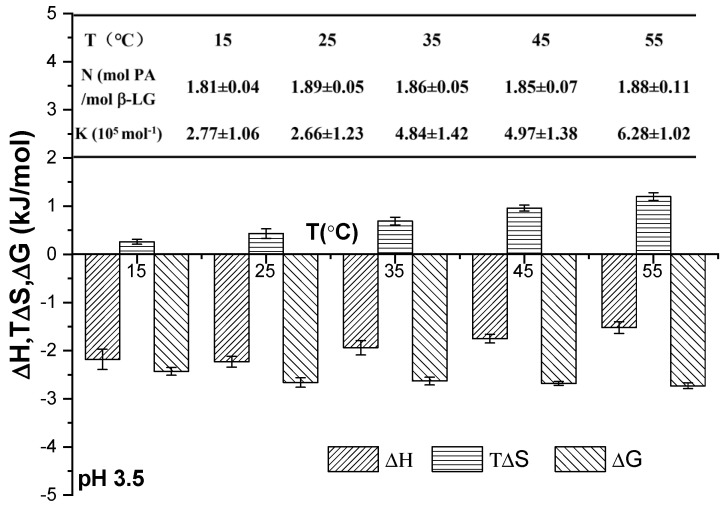
Thermodynamic parameters of binding between PA and WPI at varying temperatures.

**Figure 5 foods-13-03491-f005:**
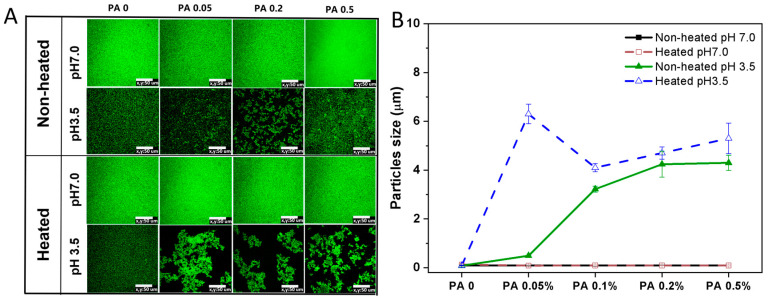
Microstructure (**A**) and particle size (**B**) of PA-WPI mixtures with varying PA (0, 0.05%, 0.1%, 0.2%, 0.5% *w*/*w*) levels before and after heat treatment at pH 7.0 and pH 3.5. The scale bar is 50 μm.

**Figure 6 foods-13-03491-f006:**
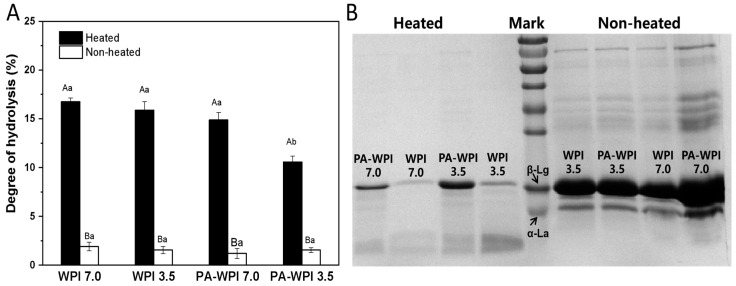
Impact of heat treatment on hydrolysis degree (**A**) and SDS-PAGE (**B**) of WPI and PA-WPI mixtures containing 0.2% *w*/*w* PA and 1.0% *w*/*w* WPI. Different capital letters (A, B) indicate significant difference between heated and non-heated samples. Different lower-case letters (a, b, c) denote significant differences between WPI and PA-WPI (LSD, *p* < 0.05).

**Figure 7 foods-13-03491-f007:**
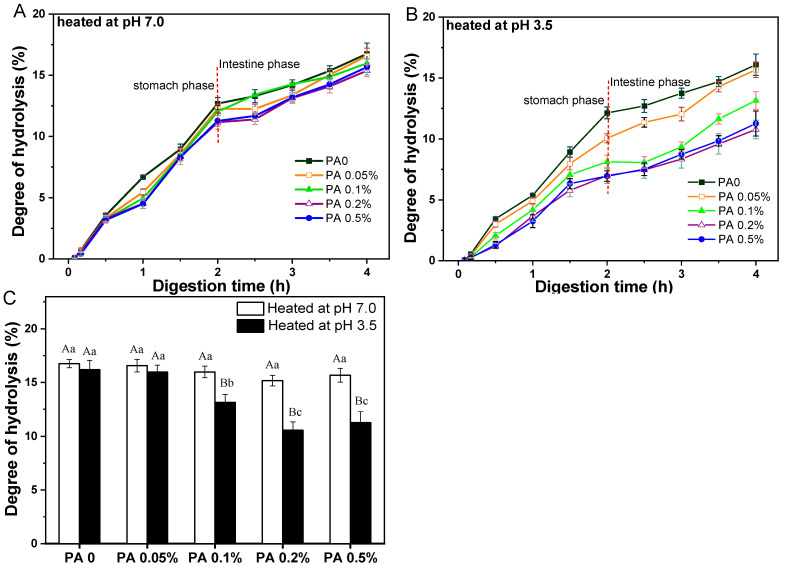
Hydrolysis degree of protein within PA-WPI mixtures containing 1.0% *w*/*w* WPI and different PA levels (0, 0.05%, 0.1%, 0.2%, 0.5% *w*/*w*) at pH 7.0 (**A**), pH 3.5 (**B**), and the final degree of hydrolysis after digestion (**C**). Different capital letters (A, B) indicate significant difference (LSD, *p* < 0.05) between samples heated at pH 7.0 and pH 3.5. Different lower-case letters (a, b) mean significant differences (LSD, *p* < 0.05) between different phytic acid levels .

**Figure 8 foods-13-03491-f008:**
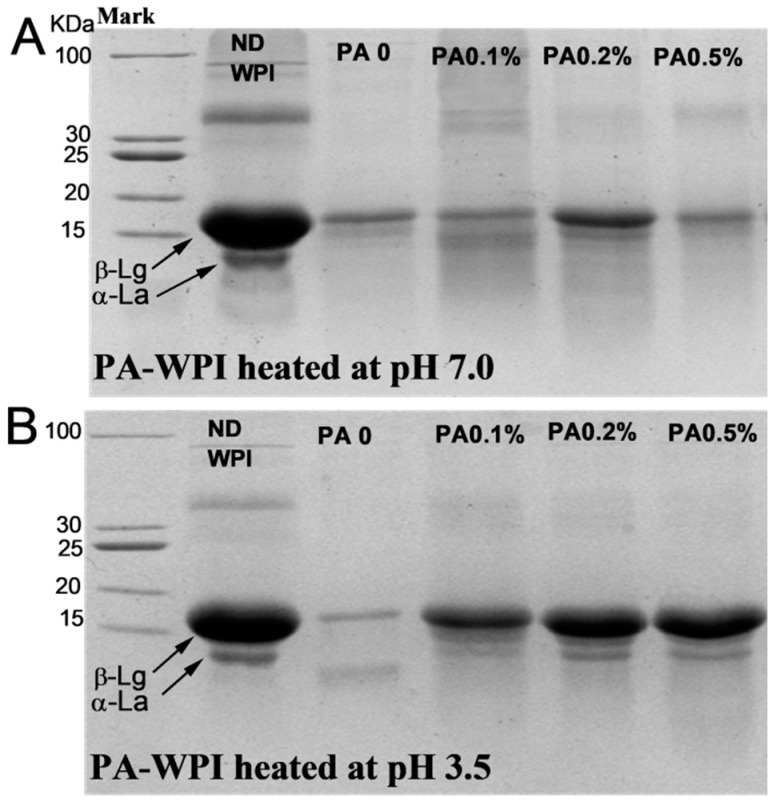
SDS-PAGE of WPI and PA-WPI mixtures containing 1.0% *w*/*w* WPI and different PA levels (0, 0.05%, 0.1%, 0.2%, 0.5% *w*/*w*) at the end of digestion: heated at pH 7.0 (**A**), heated at pH 3.5 (**B**). ND WPI represented the undigested WPI molecule.

## Data Availability

The original contributions presented in the study are included in the article, further inquiries can be directed to the corresponding author.

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
