# Peer review of "Regulation on Aggregation Behavior and In Vitro Digestibility of Phytic Acid–Whey Protein Isolate Complexes: Effects of Heating, pH and Phytic Acid Levels"

_foods, 2024, doi:10.3390/foods13213491_

Round 1
Reviewer 1 Report
Comments and Suggestions for Authors
This study is well-planned and the manuscript is written in a clear and logical way. However, there are comments need to be addressed:1. It is highly recommended to draw the chemical structures of phytic acid and WP so readers can imagine the discussion related to chemical structures. It is also recommended to show the structure of phytic acid in acidic and alkaline mediums.
2. How authors are sure that the commercial Whey Protein does not contain any metal additives/impurities? pls. explain. How might this affect the results?
3. The curve in Figure 1 reads on y axis as (100 % transmittance). Is it correct? When the turbidity is maximum the % T should be close to zero! Pls. verify and correct accordingly.
4. It is highly recommended to correlate the results obtained in figure 1 a and b with the isoelectric point of the protein.
5. In figure 3, what if the pH is low e,g, pH = 1.0
6. Please define and clarify the difference between β-Lg and β-LG
7. In figure 4, pls. explain how the free energy increased as a negative value, while the -TdS increased? Note that the enthalpy is almost constant at all temperatures.
Author Response
Comments and Suggestions for Authors
This study is well-planned and the manuscript is written in a clear and logical way. However, there are comments need to be addressed:
- Response:We thank the reviewer’s review and recognition of our work.
The language and some experimental details and analysis of the results are re-completed in the revised manuscript.
- It is highly recommended to draw the chemical structures of phytic acid and WP so readers can imagine the discussion related to chemical structures. It is also recommended to show the structure of phytic acid in acidic and alkaline mediums.
- Response:We thank the reviewer’s suggestion. The structure of phytic acid is straightforward, and the corresponding structural diagram can be found in the referenced literature[6]. Additionally, we have provided our own crafted structural diagrams that can be incorporated into the manuscript or supplementary materials as deemed necessary by reviewers and editors.
[6]Nassar, M.; Nassar, R.; Maki, H.; Al-Yagoob, A.; Hachim, M.; Senok, A.; Williams, D.; Hiraishi, N. Phytic Acid: Properties and Potential Applications in Dentistry. Frontiers in Materials 2021, 8, doi:10.3389/fmats.2021.638909
Figures of chemical structure of phytic acid and interaction mechanism of phytic acid between protein below the isoelectric point of the protein has attached in the word file.
- How authors are sure that the commercial Whey Protein does not contain any metal additives/impurities? pls. explain. How might this affect the results?
- Response:This comment goes straight to the heart of the matter. We thank the reviewer for this constructive comment. In the presence of metal ions, ternary complexes are formed between PA, WPI, and metal ions above the isoelectric point of WPI. Despite 97.6% purity, We cannot be sure that WPI does not contain any metal ions. A detailed study of the impact of metal ions and their level on the gastrointestinal fate of protein containing phytic acid are currently being carried out and will be published in another manuscript.
- The curve in Figure 1 reads on y axis as (100 % transmittance). Is it correct? When the turbidity is maximum the % T should be close to zero! Pls. verify and correct accordingly.
- Response:The pH-turbidity curve was obtained by titrating HCl (0.5, 1 M) or NaOH (0.1, 0.5 M) to adjust the pH of the PA-WPI mixed solution (from pH 7.0 to pH 2.0) while monitoring its transmittance (T%) using a UV-spectrophotometer at a wavelength of 600 nm. The turbidity of the solution was expressed as 100-T%. So the y axis is as 100-T%.
4.It is highly recommended to correlate the results obtained in figure 1 a and b with the isoelectric point of the protein.
- Response: Thanks for reviewer’s suggestions and reminders. We have identified an error in the figure1B's X-axis scale and have made corresponding modifications in the revised manuscript.
5.In figure 3, what if the pH is low e,g, pH = 1.0
- Response: As the deprotonation and protonation of PA at different pH(Crea F, De Stefano C, Milea D, et al. Formation and stability of phytate complexes in solution. Coordination Chemistry Reviews, 2008, 252(10-11): 1108-1120.), interaction of PA and WPI was weaken or not occurred due to some protonation of the phosphate groups occurred under acidic conditions. Our previous work showed that PA has almost no charge at very low pH (PEI Y, WAN J, YOU M, et al. Impact of whey protein complexation with phytic acid on its emulsification and stabilization properties. Food Hydrocolloids, 2019, 87: 90-6.), Data figure has attached in the Word file.
6.Please define and clarify the difference between β-Lg and β-LG
- Response:As suggested, we have check through the whole manuscript and unified spelling of β-Lg.
- In figure 4, pls. explain how the free energy increased as a negative value, while the -TdS increased? Note that the enthalpy is almost constant at all temperatures.
- Response:ΔH is almost constant, ΔG = ΔH - TΔS, and ΔG is negative. So as the TΔS(not -TΔS ) increased , the negative value of the free energy increases.
Reviewer 2 Report
Comments and Suggestions for Authors
The manuscript reports the results on the aggregation and digestibility of the complex of whey protein isolate (PWI) and phytic acid (PA). The results might be useful, but I have several questions mainly on the charging and aggregation.
Please provide the product numbers of PWI and PA.
What about the purity of water?
What is the used theory to calculate zeta potential from electrophoretic mobility? From the analysis of electrophoresis of lysozyme, the counterion binding and shift of slipping plane were evaluated.
The floc strength of complex of lysozyme and humic acid was studied. By reviewing such study, the authors can discus more on the aggregation.
Line 125: Malvern zetasizer uses dynamic light scattering and provides z-averaged (intensity-weighted) hydrodynamic diameter. In this regard, d_h would be better rather than D4,3 (Sauter mean?).
Line 162: “*” would be avoided for the product.
PA would be weak acid. So, please mention about the deprotonation and protonation of PA at different pH.
Figure1: the number for pH should increase from left to right in general. I am wondering why WPI shows positive zeta potential at high pH. In my memory, most of proteins show positive zeta potential at lower pH depending on the isoelectric point. What about the purity of PWI used? The number and the label of x-axis should be placed on the bottom, not in the figure, but the 0 zeta potential line should be drawn
Figure 4: "K" should be "k". Better to use the unified font.
Figure 5: What is the length for the scale bar. Nobody cannot compare the size from the pictures and that from Fig. 5B (DLS?) with the present figure.
Line 377: sampl -> sample
Author Response
The manuscript reports the results on the aggregation and digestibility of the complex of whey protein isolate (PWI) and phytic acid (PA). The results might be useful, but I have several questions mainly on the charging and aggregation.
-Response: We thank the reviewer for their positive comments and constructive suggestions on our manuscript. We have revised the manuscript according to these comments and included a detailed list of responses below.
1.Please provide the product numbers of PWI and PA.
-Response: As suggested, the product numbers of WPI (dry basis content 97.6%; BiPro JE-099-2-420) and Phytic acid (PA, 98%; Lot #BCBR3133V) have provided in the revised manuscript.
2.What about the purity of water?
-Response:All samples were prepared using double-distilled water produced via a laboratory-grade water purification system.
3.What is the used theory to calculate zeta potential from electrophoretic mobility? From the analysis of electrophoresis of lysozyme, the counterion binding and shift of slipping plane were evaluated.
-Response: The zeta potential was calculated by Smoluchowski formula. The details of the zeta potential measurements has been added to the manuscript.
4.The floc strength of complex of lysozyme and humic acid was studied. By reviewing such study, the authors can discus more on the aggregation.
-Response:As suggested, we have added an additional discussion to the revised manuscript to make this clearer.
5.Line 125: Malvern zetasizer uses dynamic light scattering and provides z-averaged (intensity-weighted) hydrodynamic diameter. In this regard, d_h would be better rather than D4,3 (Sauter mean?).
-Response:We concur with the reviewer's assessment. While in our work, both the PA-WPI complexes and the PA-WPI complexes exhibit particle sizes exceeding a micron, thus suggesting that D4,3 may offer a more appropriate representation.
6.Line 162: “*” would be avoided for the product.
-Response:We have made the changes suggested by the reviewer in the revised manuscript.
7.PA would be weak acid. So, please mention about the deprotonation and protonation of PA at different pH.
-Response:The theory of deprotonation and protonation of PA at different pH has been discussed in the literature (Crea F, De Stefano C, Milea D, et al. Formation and stability of phytate complexes in solution. Coordination Chemistry Reviews, 2008, 252(10-11): 1108-1120.). Our previous work show that PA has almost no charge at very low pH (PEI Y, WAN J, YOU M, et al. Impact of whey protein complexation with phytic acid on its emulsification and stabilization properties. Food Hydrocolloids, 2019, 87: 90-6.). Data figure has attached in the Word file.
8.Figure1: the number for pH should increase from left to right in general. I am wondering why WPI shows positive zeta potential at high pH. In my memory, most of proteins show positive zeta potential at lower pH depending on the isoelectric point. What about the purity of PWI used? The number and the label of x-axis should be placed on the bottom, not in the figure, but the 0 zeta potential line should be drawn
-Response:The reviewer's comments are accurate. The isoelectric point of WPI is pH 5.0, exhibiting a negative charge above the isoelectric point and a positive charge below it. We apologize for any reading difficulties caused by the incorrect scale format setting on the X-axis, which has been corrected in the revised manuscript.
9.Figure 4: "K" should be "k". Better to use the unified font.
-Response: As suggested, we check the whole manuscript and uniformly employ “K”.
10.Figure 5: What is the length for the scale bar. Nobody cannot compare the size from the pictures and that from Fig. 5B (DLS?) with the present figure.
-Response: We have added scale bar (50 μm) in figure title to make this clearer. The size (D4,3) was determined using a Zetasizer Nano ZS (Malvern Instruments, UK) at 25℃.
11.Line 377: sampl -> sample
-Response: Our apologies. We have corrected it in the manuscript.
Round 2
Reviewer 1 Report
Comments and Suggestions for Authors
Authors have responded to all comments.
Author Response
We thank the reviewer again for the positive comments and constructive suggestions on our manuscript.
Reviewer 2 Report
Comments and Suggestions for Authors
In general, questions raised by reviewers could be uncertainty for potential readers. In this regard, the authors should not only reply to the reviewer but also modify main text. My questions still remain.
“3.What is the used theory to calculate zeta potential from electrophoretic mobility?
From the analysis of electrophoresis of lysozyme, the counterion binding and shift of
slipping plane were evaluated.
-Response: The zeta potential was calculated by Smoluchowski formula. The details
of the zeta potential measurements has been added to the manuscript.”
The authors should justify the availability of the Smoluchowski equation for their experimental condition.
“5.Line 125: Malvern zetasizer uses dynamic light scattering and provides z-averaged
(intensity-weighted) hydrodynamic diameter. In this regard, d_h would be better
rather than D4,3 (Sauter mean?).
-Response:We concur with the reviewer's assessment. While in our work, both the
PA-WPI complexes and the PA-WPI complexes exhibit particle sizes exceeding a
micron, thus suggesting that D4,3 may offer a more appropriate representation.”
This response and the expression “particle size diameter” are nonsense. In dynamic light scattering, one can determine the diffusion coefficient of scattering objects from the fluctuation of scattered light intensity. The diffusion coefficient is converted into fluid dynamic size using the Stokes-Einstein equation. In this sense, the size should be hydrodynamic diameter. Or did the authors use the Malvern Mastersizer which is based on laser diffraction?
“7.PA would be weak acid. So, please mention about the deprotonation and protonation of PA at different pH. -Response:The theory of deprotonation and protonation of PA at different pH has been discussed in the literature (Crea F, De Stefano C, Milea D, et al. Formation and stability of phytate complexes in solution. Coordination Chemistry Reviews, 2008, 252(10-11): 1108-1120.). Our previous work show that PA has almost no charge at very low pH (PEI Y, WAN J, YOU M, et al. Impact of whey protein complexation with phytic acid on its emulsification and stabilization properties. Food Hydrocolloids, 2019, 87: 90-6.).”
OK. Such information is needed for disucssion and should be added in main text.
“8.Figure1: the number for pH should increase from left to right in general. I am wondering why WPI shows positive zeta potential at high pH. In my memory, most of proteins show positive zeta potential at lower pH depending on the isoelectric point. What about the purity of PWI used? The number and the label of x-axis should be placed on the bottom, not in the figure, but the 0 zeta potential line should be drawn -Response:The reviewer's comments are accurate. The isoelectric point of WPI is pH 5.0, exhibiting a negative charge above the isoelectric point and a positive charge below it. We apologize for any reading difficulties caused by the incorrect scale format setting on the X-axis, which has been corrected in the revised manuscript.”
OK, I understand the previous figure was competely the careless mistake by the authors. This kind of mistake could seriously ruin the relaiablity of this paper.
“The number and the label of x-axis should be placed on the bottom, not in the figure, but the 0 zeta potential line should be drawn” is not improved. The authors’ careless mistake in the previous version would originate from this point. The figure must be modified.
“9.Figure 4: "K" should be "k". Better to use the unified font. -Response: As suggested, we check the whole manuscript and uniformly employ “K””
“k” is better for “kilo”. “K” is for “Kelvin”.
“10.Figure 5: What is the length for the scale bar. Nobody cannot compare the size from the pictures and that from Fig. 5B (DLS?) with the present figure. -Response: We have added scale bar (50 mm) in figure title to make this clearer. The size (D4,3) was determined using a Zetasizer Nano ZS (Malvern Instruments, UK) at 25℃.”
OK. The aggregate size in Fig. 5A is much larger than that in Fig. 5B. Why? Is this limitation of DLS adopted by Zetasizer Nano ZS? The explanation should be added in the main text.
Author Response
We thank the reviewer again for the positive comments and constructive suggestions on our manuscript. We have revised the manuscript according to these comments and included a detailed list of responses below.
3.What is the used theory to calculate zeta potential from electrophoretic mobility?From the analysis of electrophoresis of lysozyme, the counterion binding and shift ofslipping plane were evaluated.
-Response: The zeta potential was calculated by Smoluchowski formula. The details of the zeta potential measurements has been added to the manuscript.”
The authors should justify the availability of the Smoluchowski equation for their experimental condition.
-Response: As suggested, We have added detailed zeta potential measurement method in the revised text to make it clearer: A Zetasizer Nano ZS (Malvern Instruments, UK) was ustilized to determine the zeta potential of the PA-WPI mixed solution, following the previously method[22]. Samples were loaded into an appropriate cuvette (folded capillary zeta cell DTS1070), and the zeta potential was tested by measuring the direction and velocity of the complexes in the ap-plied electric field. The Smoluchowski equation (F(ka) 1.5) was employed to calculate the zeta potential. Measurements were conducted on three freshly prepared samples, with three readings taken per sample.
[22]Zeeb, B.; Mi-Yeon, L.; Gibis, M.; Weiss, J. Growth phenomena in biopolymer complexes composed of heated WPI and pectin. Food Hydrocolloids 2018, 74, 53-61, doi:10.1016/j.foodhyd.2017.07.026.
5.Line 125: Malvern zetasizer uses dynamic light scattering and provides z-averaged(intensity-weighted) hydrodynamic diameter. In this regard, d_h would be betterrather than D4,3 (Sauter mean?).
-Response:We concur with the reviewer's assessment. While in our work, both the PA-WPI complexes and the PA-WPI complexes exhibit particle sizes exceeding a micron, thus suggesting that D4,3 may offer a more appropriate representation.”
This response and the expression “particle size diameter” are nonsense. In dynamic light scattering, one can determine the diffusion coefficient of scattering objects from the fluctuation of scattered light intensity. The diffusion coefficient is converted into fluid dynamic size using the Stokes-Einstein equation. In this sense, the size should be hydrodynamic diameter. Or did the authors use the Malvern Mastersizer which is based on laser diffraction?
-Response: According to the previous method, a Mastersizer was employed to measure the particle size of samples ranging from 0.1 to 1,000 μm, while dynamic light scattering (DLS) was utilized for samples with particle sizes between 0.3 and 6,000 nm [23]. Initially, the Malvern Mastersizer was selected for measuring the particle size of the PA-WPI complex due to its suitability for relatively larger particles. However, potential dilution and agi-tation during testing may have caused partial disruption of flocculated PA-WPI particles, resulting in undetectable results. Consequently, the dynamic light scattering (DLS) method was employed to measure the particle size of PA-WPI complexes. The mean par-ticle size diameter was determined using a Zetasizer Nano ZS (Malvern Instruments, UK) at 25℃, calculated from the translational diffusion coefficient using the Stokes–Einstein equation.
[23]Gordon, L.; Pilosof, A.M.R. Application of High-Intensity Ultrasounds to Control the Size of Whey Proteins Particles. Food Biophysics 2010, 5, 203-210, doi:10.1007/s11483-010-9161-4.
We have revised the manuscript to make this clearer.
7.PA would be weak acid. So, please mention about the deprotonation and protonation of PA at different pH.
-Response:The theory of deprotonation and protonation of PA at different pH has been discussed in the literature (Crea F, De Stefano C, Milea D, et al. Formation and stability of phytate complexes in solution. Coordination Chemistry Reviews, 2008, 252(10-11): 1108-1120.). Our previous work show that PA has almost no charge at very low pH (PEI Y, WAN J, YOU M, et al. Impact of whey protein complexation with phytic acid on its emulsification and stabilization properties. Food Hydrocolloids, 2019, 87: 90-6.).”
- Such information is needed for disucssion and should be added in main text.
-Response:As suggested, we have added a discussion about protonation and reduction in charge magnitude at lower pH value involved in the revised manuscript.
8.Figure1: the number for pH should increase from left to right in general. I am wondering why WPI shows positive zeta potential at high pH. In my memory, most of proteins show positive zeta potential at lower pH depending on the isoelectric point. What about the purity of PWI used? The number and the label of x-axis should be placed on the bottom, not in the figure, but the 0 zeta potential line should be drawn -Response:The reviewer's comments are accurate. The isoelectric point of WPI is pH 5.0, exhibiting a negative charge above the isoelectric point and a positive charge below it. We apologize for any reading difficulties caused by the incorrect scale format setting on the X-axis, which has been corrected in the revised manuscript.”
OK, I understand the previous figure was competely the careless mistake by the authors. This kind of mistake could seriously ruin the relaiablity of this paper.
“The number and the label of x-axis should be placed on the bottom, not in the figure, but the 0 zeta potential line should be drawn” is not improved. The authors’ careless mistake in the previous version would originate from this point. The figure must be modified.
-Response:Thanks for reviewer’s suggestions, we have redrawn Figure1.
“9.Figure 4: "K" should be "k". Better to use the unified font. -Response: As suggested, we check the whole manuscript and uniformly employ “K””
“k” is better for “kilo”. “K” is for “Kelvin”.
-Response: As suggested, we have corrected it in the manuscript.
“10.Figure 5: What is the length for the scale bar. Nobody cannot compare the size from the pictures and that from Fig. 5B (DLS?) with the present figure.
-Response: We have added scale bar (50 mm) in figure title to make this clearer. The size (D4,3) was determined using a Zetasizer Nano ZS (Malvern Instruments, UK) at 25℃.”
- The aggregate size in Fig. 5A is much larger than that in Fig. 5B. Why? Is this limitation of DLS adopted by Zetasizer Nano ZS? The explanation should be added in the main text.
-Response:The rationale for selecting the dynamic light scattering method to quantify the PA-WPI complex has been elucidated previously.
“However, the size observed in the microstructure images appeared larger than the particle size depicted in Figure 5b, likely due to the PA-WPI complex formign an amorphous flocculate rather than a rigid solid particle. Additionally, the particle size measured using laser scanning confocal microscopy (LSCM) was reflective of the local field of view, whereas the results from dynamic light scattering (DLS) provided an average of the overall particle size. Although discrepancies in particle size were noted between the two testing methods, both indicated the aggregation of PA and WPI below the isoelectric point, with heat treatment further intensifying this aggregation”.
We have added the explanation in the revised manuscript to make this clearer.